# SynopticMind: An Instruction Tuning MLLM For Weather Forecasting Report Generation

## Abstract

Accurate weather forecast reporting enables individuals and communities to better plan daily activities, agricultural operations, and transportation. However, the current reporting process primarily relies on manual analysis of multi-source data, which often leads to information overload and reduced efficiency. With the rapid advancement of multimodal large language models (MLLMs), leveraging data-driven models to analyze and generate reports in the weather forecasting domain remains largely underexplored. In this work, we propose the Weather Forecasting Report (WFR) task and construct the first instruction-tuning dataset for this task, named **WFInstruct**. Based on this corpus, we develop the first model, **SynopticMind**, specialized in generating weather forecast reports. Experiments on our dataset show that **SynopticMind** surpasses leading GPT-5. In addition, we analyze the generalization ability of the model, examine the influence of different visual inputs, and evaluate the contribution of individual categories of meteorological variables. **SynopticMind** offers valuable insight for developing MLLMs specialized in weather report generation. Codes are available at `https://anonymous.4open.science/r/ICLR-SynopticMind-8829`.

## 1 Introduction

Weather forecasting provides critical support for societal resilience by delivering accurate and timely predictions of atmospheric conditions. Precise forecasts of variables such as temperature, humidity, precipitation, and cloud cover facilitate effective planning in daily operations, agriculture (Ukhurebor et al., 2022), and transportation. The current forecasting workflow, illustrated in Figure 1, begins with data acquisition from in-situ stations, satellites, and radars. These observations are integrated via data assimilation to produce an initial field. Traditionally, forecasters rely on physics-based Numerical Weather Prediction (NWP) models, which solve discretized thermodynamic and fluid dynamical equations (Nathaniel et al., 2024), to generate forecast fields. Experts then analyze observational, initial, and forecast data through collaborative discussions to issue final reports. However, this process faces significant challenges: information overload from hundreds of variables across diverse data sources limits efficiency and impedes handling of complex weather phenomena. Additionally, the incorporation of subjective judgments may lead to inconsistencies across reports. Recent advances in multimodal large language models (MLLMs) offer a promising avenue to address these issues. Their ability to interpret and integrate multi-variable imagery data can assist forecasters by reducing cognitive load and minimizing subjective biases, thereby enhancing the consistency and comprehensiveness of weather reporting.

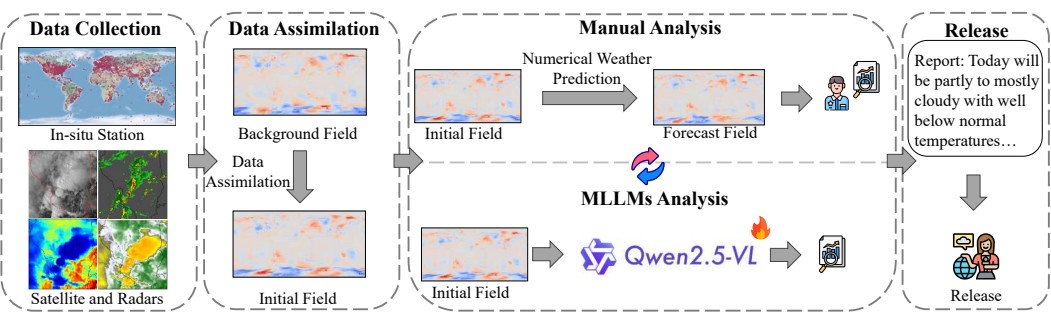

Figure 1: The pipeline of the weather forecast reporting process

MLLMs have been adopted for automated report generation in several domains. For instance, in radiology, models generate diagnostic reports from medical images like X-rays or CT scans (Li et al., 2023; Moor et al., 2023). Similarly, remote sensing applications use MLLMs to produce descriptive summaries from satellite or aerial imagery (Pang et al., 2025). These tasks demand both fine-grained visual recognition and specialized language generation. However, in weather forecasting, such applications remain underdeveloped. Despite a clear need for interpretable and accurate weather summaries, few studies have explored adapting MLLMs to generate textual reports from meteorological data—such as satellite images, radar maps, or numerical forecasts. WeatherQA (Ma et al., 2024) introduces the first multimodal dataset for severe weather report generation. However, its scope is limited to extreme events and does not cover general weather reporting. This gap stems primarily from two challenges: the scarcity of publicly available datasets with paired visual data and textual weather reports, and the limited ability of current MLLMs to perform domain-specific reasoning over complex, multi-variable meteorological inputs.

To address this gap, we firstly introduce Weather Forecasting Report (WFR) as a new task within the field of weather forecasting. Unlike the conventional pipeline, where NWP systems or foundational weather models (Lam et al., 2022; Bi et al., 2023; Pathak et al., 2022) first produce numerical outputs that are then interpreted and translated into textual reports by human forecasters, WFR is designed to directly generate human-readable weather reports from the initial atmospheric conditions at a given time $t$ and position $p$. By eliminating dependence on intermediate numerical forecasts, this approach streamlines the forecasting process and facilitates more accessible, timely information to the public.

In this paper, we construct the first Weather Forecast Report (WFR) instruction-tuning dataset, named **WFInstruct** [1], which pairs Earth Reanalysis 5 (ERA5) dataset (Hersbach et al., 2020) converted into city-level variable heatmaps as visual inputs with corresponding online weather reports as textual outputs. Based on this corpus, we propose the first open-source MLLM specialized in weather report generation. Our approach proceeds in three stages. First, we perform supervised fine-tuning (SFT) of the open-source Qwen2.5-VL-7B model using **WFInstruct**, yielding the baseline **SynopticMind** [2]. Second, to improve the lexical diversity of the generated report, we employ rejection sampling on the SFT model to create lexically diverse but semantically similar reports, which are incorporated into an augmented dataset. We then fine-tune Qwen2.5-VL on this augmented corpus, which we term Rejection Sampling Fine-Tuning (RFT). Finally, we apply Direct Preference Optimization (DPO) alignment to refine the model's style and factuality, ensuring that generated forecasts are faithful to expert descriptions while avoiding generic or inaccurate phrasing. Through extensove experiments on the **WFInstruct**, we find that

- **SynopticMind** significantly outperforms the leading closed-source MLLMs, including GPT-4o, Claude-3.7-Sonnet, and GPT-5, generating reports that are both meteorologically accurate and human-readable.
- Generalization experiments demonstrate that, when trained on data from several cities, our model generates reports for unseen cities in a zero-shot setting that still surpass the few-shot performance of GPT-4o, highlighting its strong transferability.
- We further conduct a comprehensive analysis of the effects of training data scale, the impact of different visual inputs, and the contributions of individual categories of meteorological variables. The results provide practical insights into model optimization and performance improvement.

## 2 Preliminary

**Problem Definition** Consider a weather forecast report dataset $\mathcal{D} = \{\mathbf{Q}_t^p, \mathcal{I}_t^p, \mathbf{R}_t^p\}$, where $\mathbf{Q}_t^p$ denotes the query for specific time $t$ and position $p$, with $p$ simplified to denote a city in our setting. $\mathcal{I}_t^p = \{\mathbf{I}_t^{p1}, \mathbf{I}_t^{p2}, ..., \mathbf{I}_t^{pi}\}_{i=1}^{N_v}$ represents a set of $N_v$ variable-specific weather heatmaps (e.g., temperature, precipitation, wind), which reflect meteorological conditions at time $t$ and city $p$. $\mathbf{R}_t^p$ represents the ground-truth forecast report issued at time $t$ for city $p$, which describes the predicted weather conditions for the upcoming days. The task for weather forecast report generation is to train a model that generates the forecast report $\hat{\mathbf{R}}_t^p$ conditioned on both the image set $\mathcal{I}_t^p$ and the instruction $\mathbf{Q}_t^p$.

**Visual Instruction Tuning** Visual instruction tuning has been widely used in MLLMs (Liu et al., 2023; Li et al., 2024b) to equip them with the capability to understand downstream task requirements. This approach aims to enhance the model's instruction-following ability across different

---

[1] https://huggingface.co/datasets/abcnnnnnnn/WFInstruct

[2] https://huggingface.co/abcnnnnnnn/SynopticMind

Figure 2: The overview of **SynopticMind** training framework

modalities. Multiple studies (Bai et al., 2025; Chen et al., 2024) have shown that this approach can effectively improve the zero-shot generalization abilities of MLLMs. Consider a current generative MLLM denoted as $\mathcal{M}$, which takes image set $\mathcal{I}_t^p$ and question $\mathbf{Q}_t^p$ as inputs and return a textual weather report $\hat{\mathbf{R}}_t^p$. This process can be formulated as: $\hat{\mathbf{R}}_t^p = \mathcal{M}(\mathcal{I}_t^p, \mathbf{Q}_t^p)$.

A common approach for visual instruction tuning is Supervised Fine-Tuning (SFT). The model is trained to process both the image set $\mathcal{I}_t^p$ and question $\mathbf{Q}_t^p$ as inputs and to generate textual report $\mathbf{R}_t^p$ as outputs. The SFT loss is computed based on the discrepancy between the model's predicted reports and the actual reports in the dataset, formally defined as follows:

$$\mathcal{L}_{\text{VLM}} = -\mathbb{E}_{(\mathbf{Q}_t^p, \mathcal{I}_t^p, \mathbf{R}_t^p) \sim \mathcal{D}} \left[ \log p_\theta(\mathbf{R}_t^p \mid \mathcal{I}_t^p, \mathbf{Q}_t^p) \right] \tag{1}$$

**Direct Preferred Optimization** Although visual instruction tuning can enhance the visual understanding capabilities of current MLLMs, they still struggle to generate answers that are well aligned with human preferences and intentions (Bender et al., 2021; Bommasani, 2021; Kenton et al., 2021). To bridge this gap, Reinforcement Learning from Human Feedback (RLHF) (Ouyang et al., 2022; Schulman et al., 2017; Zheng et al., 2023) has become a common approach for aligning large language models (LLMs) with human preferences. RLHF leverages human preference data to train a reward model and subsequently optimizes the policy using algorithms such as Proximal Policy Optimization (PPO) (Schulman et al., 2017). However, this method requires a reward model, which increases the training cost. To simplify the RLHF pipeline, Direct Preference Optimization (DPO) (Rafailov et al., 2023) bypasses explicit reward modeling by directly optimizing the policy using preference pairs, thereby offering improved computational efficiency and training stability. It treats the base model as a reference policy denoted as $\pi_{ref}$ and aims to optimize a new policy $\pi_\theta$ using preferred and dispreferred output pairs $(y_w, y_l)$ with logistic loss:

$$\mathcal{L}_{\text{DPO}}(\pi_\theta, \pi_{ref}) = -\mathbb{E}_{(x, y_w, y_l) \sim \mathcal{D}} \left[ \log \phi \left( \beta \log \frac{\pi_\theta(y_w|x)}{\pi_{\text{ref}}(y_w|x)} - \beta \log \frac{\pi_\theta(y_l|x)}{\pi_{\text{ref}}(y_l|x)} \right) \right], \tag{2}$$

where $\beta$ is a hyperparameter and $\phi$ is the sigmoid function. During training, DPO requires another training corpus that belongs to a domain similar to $\pi_{ref}$.

## 3 METHODOLOGY

We aim to develop a general weather report generation model capable of producing textual forecast synopses comparable to those written by experts. An overview of our framework is illustrated in Figure 2. We adopt a three-stage training strategy: (1) Supervised Fine-Tuning (SFT). In the first stage, we construct the **WFInstruct** dataset and fine-tune the base model Qwen2.5-VL-7B with it, yielding **SynopticMind**. We denote the dataset and the resulting model as $\mathcal{D}$ and $\mathcal{M}$, respectively. This stage enables the model to learn to interpret visual inputs and reason about future weather conditions, ultimately generating expert-like forecast narratives. (2) Rejection Sampling Fine-Tuning (RFT). In the second stage, we enhance the original dataset using rejection sampling to construct an augmented version, **WFInstruct-RFT**, denoted as $\mathcal{D}_{RFT}$, which promotes better alignment between visual inputs and semantically faithful yet lexically diverse descriptions. This augmented dataset is then used to fine-tune the base Qwen-2.5-VL-7B model, obtaining **SynopticMind-RFT**, denoted as $\mathcal{M}_{RFT}$. and (3) Direct Preference Optimization (DPO). In the final stage, we construct a preference dataset **WFInstruct-DPO**, denoted as $\mathcal{D}_{DPO}$, and apply Direct Preference Optimization (DPO) to further refine the model $\mathcal{M}_{RFT}$. This process yields **SynopticMind-DPO**, denoted as $\mathcal{M}_{DPO}$, which generates higher-quality forecasts that align more closely with human preferences and expert judgment. The statistics of the data used in each stage are summarized in Table 1. In the following sections, we provide detailed descriptions of the dataset construction and training strategies for each stage.

Table 1: Statistics of our corpus, including total image sets ($\mathcal{I}$) and report $\mathbf{R}$.

|  | Data | Year | hk | box | lox | lwx | mtr | okx | pbz | pqr | sew | vef | **Sum.** |
|---|---|---|---|---|---|---|---|---|---|---|---|---|---|
| **WFInstruct** | Total $\mathcal{I} - \mathbf{R}$ | 2017-2021 | 1823 | 1462 | 1812 | 1827 | 1783 | 1827 | 1814 | 1359 | 1170 | 1547 | 16424 |
| **WFInstruct-RFT** | Total $\mathcal{I}$ | 2017-2020 | 1461 | 1097 | 1448 | 1462 | 1428 | 1462 | 1455 | 1091 | 915 | 1205 | 13024 |
|  | Total $\mathbf{R}$ |  | 1847 | 3497 | 4904 | 5799 | 5095 | 6137 | 4906 | 3579 | 3262 | 4029 | 43055 |
| **WFInstruct-DPO** | Total $\mathcal{I} - (\mathcal{Y}_w, \mathcal{Y}_l)$ | 2021 | 354 | 365 | 364 | 365 | 355 | 365 | 359 | 268 | 255 | 342 | 3392 |
| **Test Set** | Total $\mathcal{I} - \mathbf{R}$ | 2022 | 365 | 364 | 361 | 364 | 349 | 365 | 344 | 289 | 253 | 347 | 3401 |

## 3.1 TRAINING STRATEGY

**Stage 1 & Stage 2** We adopt a mixed-task instruction tuning strategy, in which the model is trained simultaneously on data from 10 different cities. This approach aims to equip the model with the ability to generate diverse forecast reports tailored to various regional contexts, which is defined as follows:

$$\mathcal{L}_{\text{VLM}} = -\mathbb{E}_{(\mathbf{Q}_t^p, \mathcal{I}_t^p, \{\mathbf{R}_{t,i}^p\}_{i=1}^N) \sim \mathcal{D}} \left[ \sum_{i=1}^N \log p_\theta(\mathbf{R}_{t,i}^p \mid \mathcal{I}_t^p, \mathbf{Q}_t^p) \right]. \tag{3}$$

where $p$ denotes the cities, $t$ denotes the time, $\theta$ are the parameters of the model and $N$ denotes of the number of reports for each sample. In stage 1, $N$ is fixed to 1, whereas in stage 2, the value of $N$ depends on the candidate votes for the data corresponding to city $p$ and time $t$, as described in Section 3.2.2. The optimization process trains the model to align visual meteorological inputs with corresponding textual forecasts, guiding it to effectively follow the given instructions, yielding the model $\mathcal{M}$ and $\mathcal{M}_{RFT}$, respectively.

**Stage 3** In stage 3, we employ DPO to encourage the model to learn from positive examples that are semantically similar but lexically dissimilar to the ground truth, while negative examples are lexically similar but semantically different, thereby guiding the model to focus on semantic understanding rather than superficial word matching. This setup plays a crucial role in the context of weather report generation. Models can easily exploit lexical overlap by memorizing common phrases like "partly cloudy" or "chance of showers," without truly understanding the underlying meteorological conditions. We collect 3392 preference samples named $\mathcal{D}_{DPO}$ for our DPO training, and the training objective for DPO is defined as follows:

$$\mathcal{L}_{\text{DPO}}(\pi_\theta, \pi_{ref}) = -\mathbb{E}_{(x, \mathcal{Y}_w, \mathcal{Y}_l) \sim \mathcal{D}_{DPO}} \left[ \log \phi \big( \beta \log \frac{\pi_\theta(\mathcal{Y}_w|x)}{\pi_{\text{ref}}(\mathcal{Y}_w|x)} - \beta \log \frac{\pi_\theta(\mathcal{Y}_l|x)}{\pi_{\text{ref}}(\mathcal{Y}_l|x)} \big) \right]. \tag{4}$$

where $\pi_{\text{ref}}$ refers to the reference model $\mathcal{M}_{RFT}$, and the fully trained model $\pi_\theta$ is denoted as $\mathcal{M}_{DPO}$.

## 3.2 DATASET COLLECTION

### 3.2.1 FOUNDATION CORPUS

To achieve this goal, the main challenge lies in the lack of data. To address this challenge, we developed the first multimodal weather report instruction tuning dataset **WFInstruct**. This dataset is consist of (1) the visualizations of meteorological variable around each city in the format of heatmaps; (2) and the weather report written by experts at the corresponding time, providing forecasts for the upcoming period. This section outlines the detailed dataset curation procedure, including city report collection and visual input generation.

**Textual Data** We first collect weather reports from 10 different cities online, with details provided in Appendix E. Since multiple similar reports may be issued on the same day, we retain only one report per city per day to enhance data diversity. As expert reports often contain geographic entities (e.g., place names and regions), we explicitly annotate these entities in the following visual data to help MLLMs better understand the corresponding geographic information in the images. Specifically, we first use the English core web model (Honnibal et al., 2020) to identify and extract all geographic named entities from all the reports for each city. We then identify high-frequency entities and use them to annotate the following variable heatmaps.

**Visual Data** For the visual input, we use the ERA5 reanalysis dataset to construct regional variable heatmaps. Firstly, we select 12 single-level variables; the detailed information is shown in Table 8 in the Appendix. The dataset consists of hourly data with a spatial resolution of $0.25°$. Rather than utilizing global ERA5 data directly, we extract regional data corresponding to the geographic region of each city to better align with our task. This localized approach ensures that the visual inputs

accurately reflect the meteorological conditions relevant to each target city, thereby improving the relevance of the generated reports. We curate our foundation training set using data from 2017 to 2021, and reserve the 2022 data as the testing set.

### 3.2.2 DATA AUGMENTATION

Due to the limited availability of textual data, lexically diverse yet factually consistent descriptions are scarce. Lexical diversity is important, as it enables the model to capture the underlying factual phenomena of weather rather than relying on superficial wording. For instance, the term "low pressure" is often associated with "unsettled weather conditions", both pointing to the same underlying phenomenon. Therefore, we adopt a rejection sampling strategy to improve the lexical diversity of the generated reports. Specifically, we performed inference 50 times on a subset of $\mathcal{D}$ covering the years 2017–2020 using the $\mathcal{M}$ trained in Stage 1 with a temperature setting of 0.9, while reserving the 2021 subset for DPO training in the next stage. We are then required to select the most accurate generated weather report. However, since the generated reports are open-ended, traditional evaluation metrics based on lexical or statistical overlap with the ground truth are limited in their ability to assess the overall quality and informativeness of the outputs. Thus, we adopt the LLM-as-a-Judge approach to assess the *Correctness* of the reports. Specifically, we use GPT-4o to score the generated reports on a scale from 0 to 5, based on their alignment with the ground truth reports. The prompt details can be found in the Appendix Table13.

**Diverse Report Selection**   Since not every generated sample yields a report with the maximum score of 5, we set different thresholds for different cities: specifically, a threshold of 5 for HK, and a threshold of 4 for the other cities. For each sample, we identify generated reports whose score meets or exceeds the corresponding city-specific threshold as the reference reports. Based on this, we aim to select a subset of lexically diverse reports from these reference reports, ensuring both factual accuracy and variation in expression. We assume that reference reports exhibiting the greatest distance from the ground truth report can be considered as diverse candidates. Based on this assumption, we adopt four distance-based strategies to measure diversity: (1) edit distance, (2) TF-IDF similarity(Bafna et al., 2016), (3) Jaccard similarity, and (4) cosine similarity computed using Sentence-BERT(Reimers & Gurevych, 2019) embeddings. As it is possible for different strategies to select the same candidate, we retain each unique candidate only once to prevent overfitting.

We incorporate the selected candidate reports into $\mathcal{D}$ (2017-2020 subset). The resulting dataset constitutes our final $\mathcal{D}_{RFT}$ . $\mathcal{D}_{RFT}$ not only provides high-quality reports with strong correctness, but also introduces rich linguistic diversity for each visual input. For instance, "a weak cold front will arrive" and "temperatures will dip slightly due to a frontal system" convey the same information using different expressions. Although weather reports often describe the same meteorological phenomena, they can differ significantly in phrasing. This diversity plays a crucial role in enhancing the image-text alignment capabilities of MLLMs.

### 3.2.3 PREFERENCE DATA SELECTION

During training stage 3, we employ Direct Preference Optimization (DPO) to encourage the model to generate higher-quality forecasts. DPO is applied to the RFT-trained model $\mathcal{M}_{RFT}$, further aligning it with human-preferred outputs using a training corpus drawn from a similar domain as the SFT data. Specifically, DPO leverages input pairs labeled as $(\mathcal{Y}_w, \mathcal{Y}_l)$, where $\mathcal{Y}_w$ and $\mathcal{Y}_l$ represent the preferred and less preferred reports, respectively.

Instead of directly using the ground-truth reports as the preferred outputs $\mathcal{Y}_w$, we construct preference pairs $(\mathcal{Y}_w, \mathcal{Y}_l)$ from the generated candidates. Specifically, We use $\mathcal{M}_{RFT}$ to generate 50 candidate reports for each input from a subset of $\mathcal{D}$ collected in 2021. We employ the same LLM-as-a-judge strategy as described in Section 3.2.2. For each sample, we select the highest-scoring reports among the generated reports as the preferred candidates $\mathcal{Y}_w^C$, while the second highest-scoring reports are treated as less preferred candidates $\mathcal{Y}_l^C$. Similarly, we adopt four strategies to measure the semantic similarity between the candidate and ground-truth reports, following the approach in Section 3.2.2. Among the candidates $\mathcal{Y}_w^C$, we select the report $\mathcal{Y}_w^P$ that is most dissimilar to the ground truth. Similarly, among the candidates $\mathcal{Y}_l^C$, we select the report $\mathcal{Y}_l^P$ that is most similar to the ground truth. A majority voting strategy is then applied to determine the most frequently selected reports from $\mathcal{Y}_w^P$ and $\mathcal{Y}_l^P$ across the four similarity methods, resulting in the final preference pair $(\mathcal{Y}_w, \mathcal{Y}_l)$.

## 4 EXPERIMENTS

**Baselines**   We evaluate state-of-the-art MLLMs, GPT-4o, Gemini-1.5-pro, Claude-3.7-Sonnet, Gemini-2.5-Pro and GPT-5, on our weather report dataset. All models are configured with a temper-

Table 2: Performances of **SynopticMind** and other baselines on **WFInstruct** test set.

| Algorithms | HK | BOX | LOX | LWX | MTR | OKX | PBZ | PQR | SEW | VEF | Average |
|---|---|---|---|---|---|---|---|---|---|---|---|
| *Closed-source MLLMs* | | | | | *Correctness* | | | | | | |
| GPT-4o (zero-shot) | 2.51 | 2.05 | 1.84 | 1.46 | 2.03 | 1.72 | 2.04 | 1.92 | 1.77 | 1.84 | 1.92 |
| GPT-4o (3-shot) | 3.29 | 2.63 | 2.68 | 2.52 | 2.75 | 2.63 | 2.59 | 2.69 | 2.49 | 2.62 | 2.69 |
| Gemini-1.5-Pro (3-shot) | 3.26 | 2.83 | 2.72 | 2.94 | 2.70 | 3.28 | 2.53 | 2.93 | 2.86 | 2.70 | 2.87 |
| Claude-3.7-Sonnet (3-shot) | 2.75 | 2.46 | 2.28 | 2.20 | 2.32 | 2.68 | 2.04 | 2.60 | 2.00 | 2.22 | 2.35 |
| Gemini-2.5-Pro (zero-shot)* | 2.76 | 2.58 | 2.38 | 2.27 | 2.61 | 2.27 | 2.53 | 2.74 | 2.68 | 2.74 | 2.55 |
| GPT-5 (3-shot) | 3.62 | 3.17 | 3.11 | 2.98 | 3.21 | 3.45 | 2.98 | 3.22 | 3.07 | 3.27 | 3.20 |
| *Open-source MLLMs* | | | | | | | | | | | |
| **SynopticMind** | 3.66 | 2.79 | 2.58 | 3.05 | 2.81 | 3.28 | 2.85 | 2.73 | 2.93 | 2.63 | 2.93 |
| **SynopticMind-RFT** | 3.70 | 2.90 | 3.14 | 3.31 | 3.03 | 3.54 | 3.02 | 3.16 | 3.43 | 2.90 | 3.21 |
| **SynopticMind-DPO** | 3.47 | 3.28 | 3.48 | 3.41 | 3.53 | 3.81 | 3.13 | 3.54 | 3.74 | 3.40 | 3.48 |
| *Closed-source MLLMs* | | | | | *Detailedness* | | | | | | |
| GPT-4o (zero-shot) | 2.73 | 1.85 | 1.83 | 1.76 | 2.09 | 1.83 | 2.35 | 1.95 | 1.77 | 1.96 | 2.01 |
| GPT-4o (3-shot) | 3.50 | 2.56 | 2.77 | 2.93 | 2.93 | 2.83 | 2.82 | 2.85 | 2.79 | 2.66 | 2.86 |
| Gemini-1.5-Pro (3-shot) | 3.41 | 3.16 | 3.09 | 3.16 | 3.07 | 3.30 | 2.86 | 3.17 | 3.21 | 3.0 | 3.14 |
| Claude-3.7-Sonnet (3-shot) | 3.24 | 2.56 | 2.54 | 2.58 | 2.67 | 2.95 | 2.59 | 2.67 | 2.45 | 2.57 | 2.68 |
| Gemini-2.5-Pro (zero-shot)* | 3.56 | 2.97 | 3.15 | 2.82 | 3.37 | 2.83 | 3.04 | 3.32 | 3.15 | 2.98 | 3.11 |
| GPT-5 (3-shot) | 3.75 | 3.46 | 3.58 | 3.49 | 3.50 | 3.78 | 3.52 | 3.44 | 3.35 | 3.48 | 3.53 |
| *Open-source MLLMs* | | | | | | | | | | | |
| **SynopticMind** | 3.81 | 2.71 | 2.75 | 3.41 | 3.10 | 3.46 | 3.09 | 2.93 | 3.23 | 2.97 | 3.15 |
| **SynopticMind-RFT** | 3.81 | 3.09 | 3.47 | 3.62 | 3.38 | 3.67 | 3.54 | 3.48 | 3.71 | 3.26 | 3.46 |
| **SynopticMind-DPO** | 3.47 | 3.65 | 3.98 | 3.89 | 4.04 | 3.99 | 3.88 | 3.91 | 4.07 | 3.86 | 3.87 |

ature setting of 0.1. Following the prompt design in previous work (Ma et al., 2024), the zero-shot setting includes meteorological variable visual inputs, corresponding explanations for each parameter, and the timestamp of the data to ensure comprehensive context. Additionally, step-by-step analysis instructions are provided to guide the model through the figures in order to generate the weather report. To improve the quality of the few-shot examples, we select them from the corresponding day of previous years (2017–2021) for each test sample. Further prompt details are provided in the Appendix Table11 and Table12.

**Metric** Given the open-ended nature of report generation, we follow evaluation strategies from video captioning (Cheng et al., 2024), utilizing LLM-as-a-Judge methods to evaluate the generated reports against the ground truth. Specifically, we use GPT-4o to score each report a score from 0 to 5 based on two key criteria: *Correctness* of Information, which evaluates whether the content aligns with the human-written report and avoids misinterpretation or misinformation; and *Detailedness* of Orientation, which assesses the completeness of the response, focusing on specific and comprehensive details rather than generic statements. Further prompt details are provided in the Appendix Table13 and Table14.

**Implementation** We employ the open-source Qwen2.5-VL-7B as our backbone model, and our implementation is built on DeepSpeed, HuggingFace, and the LLaMA Factory library. In both the SFT and RFT stages, we set the learning rates for the LLM, merger, and vision encoder to 1e-5, 1e-5, and 2e-6, respectively. The model is trained for one epoch with a per-GPU batch size of 4 on 4 NVIDIA A800 GPUs. During the DPO stage, the learning rate is reduced to 5e-7, the vision tower is frozen, and the per-GPU batch size is set to 1, with the $\beta$ hyperparameter fixed at 0.1. For evaluation, we use a temperature of 0 and limit the maximum output length to 200 tokens, ensuring stable and reliable generations.

### 4.1 Overall Performance

We present the results of the evaluation on the test set across ten cities in Table 2. Based on these results, we have the following observations:

- Our models consistently outperform all leading closed-source MLLMs in all tasks. RFT yields noticeable improvements in both *Correctness* and *Detailedness* across each city, which demonstrates the effectiveness of data augmentation. DPO further enhances the average performance across tasks. However, DPO adversely affects the HK task. This is because HK reports focus on short-term forecasting, which is an easier task than the short-to-medium-term reports required for

---

[2]* API currently can process a maximum of 16 images, limiting our test to the 0-shot setting.

American cities. Since RFT already achieves strong performance here, further application of DPO results in degradation, which demonstrates that it requires further tuning.

- Despite overall improvements across the board, certain regions remain more challenging for report generation. For example, in PBZ, the *Correctness* scores for our models are 2.85, 3.02, and 3.14, respectively, which is noticeably below the average. It indicates that these locations present persistent difficulties for the models, suggesting that the weather conditions in these regions are more complex, increasing the learning difficulty for the model.

### 4.2 DATA VOLUME ANALYSIS

To further assess the influence of data volume and diversity on model performance, we examine the effect of the maximum retained reports per case during RFT. The *Correctness* results are shown in Figure 3. We observe a consistent improvement in model performance as the training corpus expands. The most substantial gains occur when the reports grows from size 1 to 2, while subsequent expansions yield diminishing marginal improvements. This trend suggests that increased diversity in the reports generally enhances the model's ability to align meteorological visual inputs with textual outputs.

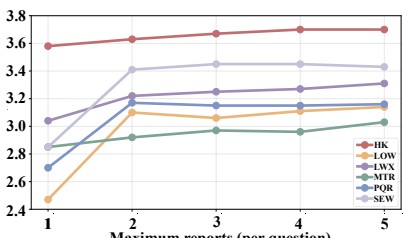

Figure 3: Performances with increasing number of reports for each question

### 4.3 GENERALIZATION ANALYSIS

To further assess the generalization capability of the model after RFT, we group the U.S. cities into three geographic regions: Northwest (SEW, PQR), Southwest (LOX, MTR, VEF), and Northeast (PBZ, LWX, BOX, OKX). We then train two specialized models to evaluate generalization under geographically close and distant conditions. *Correctness* results are presented in Figure 4.

**Geographically Close Generalization** We train a model, designated as **SynopticMind-General**, using data from cities (PBZ, LWX, MTR, VEF, SEW) and evaluate it on unseen geographically close cities (OKX, LOX, PQR, BOX). Under a zero-shot setting, it consistently outperforms GPT-4o in zero-shot evaluation and even exceeds GPT-4o's three-shot performance for cities such as OKX and PQR. These results demonstrate the model's strong generalization ability within geographically coherent regions.

**Geographically Distant Generalization** The model **SynopticMind-Regional** is trained on data from northeastern cities, and tested on cities from other regions. In this challenging scenario, our model still outperforms the zero-shot setting of GPT-4o, although it does not surpass the three-shot setting. This is primarily due to the significant geographic differences, which result in substantial variations in weather patterns and consequently hinder the model's generalization ability.

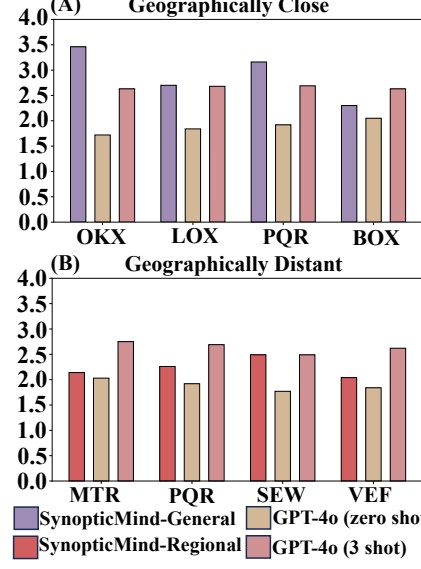

Figure 4: Generalization analysis

### 4.4 VISUAL INPUT ANALYSIS

To investigate how different types of meteorological visual inputs affect SFT model performance, we construct two extended datasets based on the default regional single-level **WFInstruct**.

First, to capture multi-level atmospheric information beyond single-level variables, we introduce **WFInstruct-Pressure**, which augments the base data with pressure-level variables. Second, to assess the effect of the broader geographical context, we build **WFInstruct-Global** by incorporating larger-scale regional data. The detailed configuration is shown in Appendix E.3. We fine-tune the model under each visual input configuration during SFT and report the *Correctness* results on their respective test sets in Table 3.

As shown in Table 3, all three settings achieve the same average correctness score of 2.93, indicating comparable overall performance. However, detailed city-level comparisons reveal nuanced differences. For instance, the model trained with **WFInstruct-Pressure** performs best in cities such as

Table 3: The impact of different visual inputs on training

| Dataset | HK | BOX | LOX | LWX | MTR | OKX | PBZ | PQR | SEW | VEF | Average |
|---|---|---|---|---|---|---|---|---|---|---|---|
| **WFInstruct-Pressure** | 3.67 | 2.59 | 2.59 | 3.04 | 2.83 | 3.34 | 2.86 | 2.92 | 2.84 | 2.60 | 2.93 |
| **WFInstruct-Global** | 3.59 | 2.58 | 2.72 | 3.11 | 2.80 | 3.32 | 2.89 | 2.86 | 2.85 | 2.59 | 2.93 |
| **WFInstruct** | 3.66 | 2.79 | 2.58 | 3.05 | 2.81 | 3.28 | 2.85 | 2.73 | 2.93 | 2.63 | 2.93 |

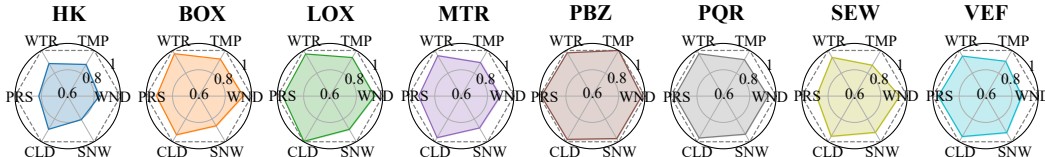

Figure 5: The impact of different categories of variables on training

MTR and PQR, suggesting that the inclusion of vertical atmospheric structure can benefit forecasting in certain regions. In contrast, the default setting achieves the highest scores in BOX, SEW and VEF, implying that simpler, localized visual features may suffice for these areas.

Model trained with **WFInstruct-Global**, which incorporates broader regional context, performs best in cities like HK and PBZ, but lags behind in BOX and SEW. This suggests that large-scale geographic inputs may introduce useful information for regions with sparse local signals, yet may also lead to noise or distribution mismatch in areas with strong local dynamics. Overall, these results highlight that while the three visual input strategies yield similar average correctness, the optimal input configuration may vary across locations. In fact, the visual input is not limited to the initial state of the variable. The NWP prediction variable can be further incorporated to augment the visual information. The results are shown in Appendix Table 4.

### 4.5 VARIABLE ANALYSIS

To evaluate the contribution of individual meteorological variables to forecasting performance, we perform ablation studies by providing only one category of variables as input at a time. This setup assesses the predictive information contained in each variable class when used in isolation. The variables are grouped into six categories: wind (WND), temperature (TMP), water-related features (WTR), surface pressure (PRS), cloud cover (CLD), and snow depth (SNW); see Appendix Table 7 for details. For each category, we SFT a separate model using only variables from that group. Performance is normalized relative to the full-variable model (**SynopticMind**), which serves as the upper bound. Results are presented in Figure 5.

We observe that using snow as an independent variable performs farthest from the SFT baseline, indicating that snow contributes fewer predictive signals. In contrast, other variables play a more important role in weather report generation. Notably, each city exhibits unique patterns. In Hong Kong (HK), single-variable models perform markedly worse than in U.S. cities, which we attribute to differences in forecast horizons: HK reports focus on short-term forecasts that rely heavily on immediate inputs, while U.S. forecasts include medium-term predictions, diluting the dependence on individual variables. We also find that variable sensitivity varies regionally. In PBZ, for instance, all single-variable models perform near the SFT baseline, suggesting forecast accuracy depends less on specific variables and more on historical temporal context. This implies the model may rely on learned periodic patterns rather than real-time meteorological inputs in certain regions.

### 4.6 CASE STUDY

We present a case study of the weather report issued at local time 08:00 on May 27, 2022 (Friday) in New York. As shown in Figure6, the ground truth forecast can be broken down into four distinct meteorological phases: (1) A cold front approaches today (Friday), (2) The cold front passes through on Saturday, (3) High pressure dominates the region through Wednesday of the following week, and (4) A potential frontal boundary may impact the area on Thursday next week.

The report generated by **SynopticMind** accurately identifies the evolution of these events, including the initial stalled front, its northward movement on Saturday, the establishment of high pressure early next week, and the possibility of another frontal system by midweek. In contrast, GPT-4o (3-shot) fails to capture the multi-stage temporal evolution, including the Saturday frontal passage

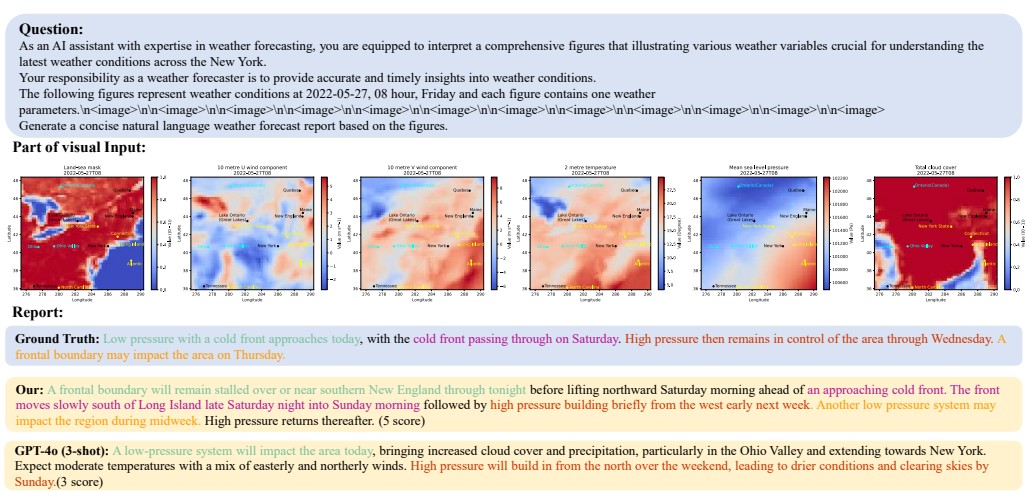

**Question:**
As an AI assistant with expertise in weather forecasting, you are equipped to interpret a comprehensive figures that illustrating various weather variables crucial for understanding the latest weather conditions across the New York.
Your responsibility as a weather forecaster is to provide accurate and timely insights into weather conditions.
The following figures represent weather conditions at 2022-05-27, 08 hour, Friday and each figure contains one weather parameters.\n<image>\n\n<image>\n\n<image>\n\n<image>\n\n<image>\n\n<image>\n\n<image>\n\n<image>\n\n<image>\n\n<image>
Generate a concise natural language weather forecast report based on the figures.

**Part of visual Input:**

**Report:**

**Ground Truth:** Low pressure with a cold front approaches today, with the cold front passing through on Saturday. High pressure then remains in control of the area through Wednesday. A frontal boundary may impact the area on Thursday.

**Our:** A frontal boundary will remain stalled over or near southern New England through tonight before lifting northward Saturday morning ahead of an approaching cold front. The front moves slowly south of Long Island late Saturday night into Sunday morning followed by high pressure building briefly from the west early next week. Another low pressure system may impact the region during midweek. High pressure returns thereafter. (5 score)

**GPT-4o (3-shot):** A low-pressure system will impact the area today, bringing increased cloud cover and precipitation, particularly in the Ohio Valley and extending towards New York. Expect moderate temperatures with a mix of easterly and northerly winds. High pressure will build in from the north over the weekend, leading to drier conditions and clearing skies by Sunday.(3 score)

Figure 6: A typical case of **SynopticMind** and GPT-4o.

and the broader high-pressure and frontal developments later in the week. This case demonstrates **SynopticMind**'s superior capability in meteorological reasoning and sequential event modeling.

## 5 RELATED WORKS

**Domain-Specific Report Generation**  Multimodal large language models (MLLMs) are increasingly applied to automated report generation in domain-specific tasks that demand structured and context-aware summaries. In radiology, for instance, MLLMs generate diagnostic reports directly from medical images such as X-rays and CT scans (Li et al., 2023; Moor et al., 2023). Similarly, remote sensing leverages MLLMs to produce textual descriptions from satellite and aerial imagery (Pang et al., 2025; Kuckreja et al., 2024; Zhang et al., 2024), supporting applications including land use monitoring, disaster assessment, and environmental analysis. Despite these advancements, the use of MLLMs for weather report generation which requires interpreting meteorological signals to produce public-facing forecasts remains underexplored.

**Earth VLMs**  To capture the alignment between variables and textual descriptions, OmniEarth-Bench (Wang et al., 2025) introduced a multimodal Earth science dataset to benchmark existing MLLMs across multiple dimensions. However, it primarily focuses on Earth science knowledge questions in a multiple-choice format. Similarly, CLLMate (Li et al., 2024a) proposed a multimodal dataset for reasoning about weather parameters and predicting severe weather in real-world scenarios, formulating the problem as a multiple-choice task through a hierarchical categorization of weather and climate events. WeatherQA (Ma et al., 2024) further enables summary generation but remains centered on severe weather event prediction. These works exhibit two notable limitations: (1) their multiple-choice design, while suitable for classification-style evaluation, is inherently limited in assessing free-form, context-rich weather report generation, and thus falls short of supporting general forecasting report; and (2) their focus is largely confined to severe weather events, leaving the broader scope of everyday, fine-grained weather reporting underexplored.

## 6 CONCLUSION

In this work, we propose WFR task, which aims to generate comprehensive textual weather prediction reports based on initial weather conditions. To address data scarcity in this domain, we construct **WFInstruct**, the first instruction-tuning multimodal dataset encompassing both weather conditions and textual reports across 10 cities. We further fine-tune the open-source MLLM Qwen2.5-VL-7B, resulting in **SynopticMind**. Our model not only surpasses GPT-5 in performance but also exhibits strong cross-city transferability for weather report generation. Experiments show that increasing training volume during RFT effectively boosts model performance, while direct preference optimization further enhances report quality. In future work, we plan to incorporate data from additional cities and train a global report model with strong generalization capability.

## 7 REPRODUCIBILITY STATEMENT

The data processing steps and training pipeline are available at https://anonymous.4open.science/r/ICLR-SynopticMind-8829. The dataset **WFInstruct** and the model **SynopticMind** can be accessed at https://huggingface.co/datasets/abcnnnnnnn/WFInstruct and https://huggingface.co/abcnnnnnnn/SynopticMind, respectively.

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

## A  THE USE OF LLMS

In this work, we used LLMs as an auxiliary tool for polishing the writing and improving the clarity of the manuscript. Besides, we employed LLMs as evaluators. Specifically, the LLM was used to assign scores to generated reports by comparing them with the ground-truth references.

## B  ADDITIONAL RELATED WORK

**Weather Prediction**  Numerical Weather Prediction (NWP) relies on the equations of thermodynamics and fluid dynamics to model the dynamic interactions among the atmosphere, land, and ocean systems. With the advancement of deep learning, numerous studies have explored the use of radar imagery (Veillette et al., 2020) for precipitation nowcasting (Wen et al., 2024; Gao et al., 2023), while Digital Typhoon (Kitamoto et al., 2023) leverages typhoon satellite imagery for spatiotemporal forecasting. However, these works often focus on a single meteorological variable and lack accompanying textual event narratives. The release of the ECMWF Reanalysis v5 (ERA5) dataset (Hersbach et al., 2020) has further enabled the development of weather foundation models such as FourcastNet (Pathak et al., 2022), GraphCast (Lam et al., 2022), FuXi (Chen et al., 2023), and Pangu-Weather (Bi et al., 2023). Nevertheless, these models primarily focus on predicting numerical meteorological variables rather than directly producing textual forecasts for public communication.

## C  LIMITATION

Due to the scarcity of open-source weather forecast report data, our experiments primarily focus on Hong Kong and selected cities in the United States. In future work, we plan to collect reports from more countries and train models that generalize to broader regions. In addition, our current approach only uses the initial state of the weather as input for simplicity. Extending the model to incorporate multiple prediction steps from HRES as visual inputs could provide richer information and further improve performance.

## D  ADDITIONAL EXPERIMENTS

### D.1  DATA VOLUME ANALYSIS

The *Detailness* results are shown in Table 7. We observe a consistent improvement in model performance as the training corpus expands. This trend indicates that greater diversity in the reports generally enhances the model's ability to align meteorological visual inputs with textual outputs.

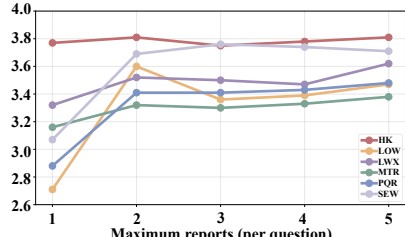

Figure 7: Performances with increasing number of reports for each question in RFT

However, in certain cases such as the city LOX, the *Detailedness* score drops from 3.6 to 3.3 when the number of reports increases from 2 to 3. This decline potentially indicates overfitting, where the model becomes overly specialized in the training data patterns, reducing its ability to generalize to new scenarios.

### D.2  TEMPORAL ANALYSIS

To investigate whether model performance is affected by seasonal variations, We further divided the test set results into four quarters corresponding to the months (Q1: January–March, Q2: April–June, Q3: July–September, Q4: October–December). Based on this division, we compared the performance across the two evaluation dimensions, correctness and detailedness, for ten representative cities. As shown in Figure 8, $\mathcal{D}_{DPO}$ consistently achieves the highest scores across quarters and regions, highlighting its robustness against seasonal variations. In contrast, GPT-4o (3 shot) performs the weakest, while $\mathcal{D}$ and $\mathcal{D}_{RFT}$ lie in between, with the latter showing slightly improved stability. These results demonstrate that our domain-specific fine-tuning strategy enables SynopticMind

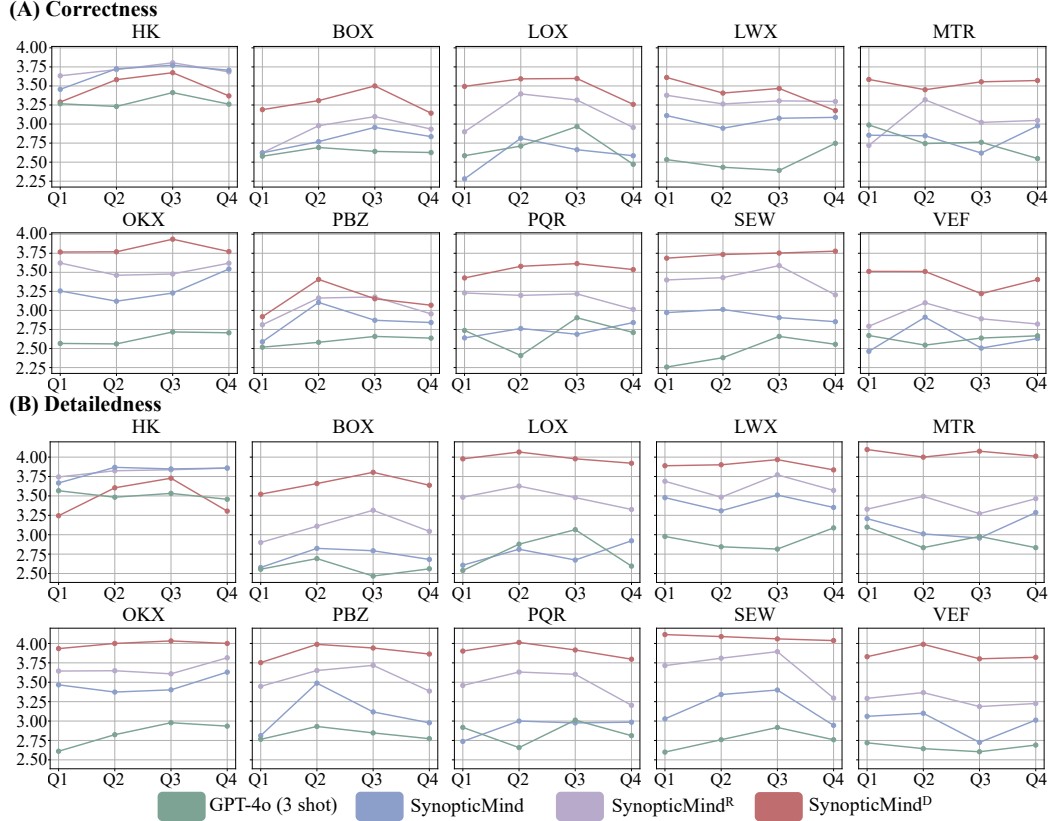

Figure 8: Comparison of quarterly average scores across ten cities for two evaluation dimensions: (a) correctness and (b) detailedness.

to better capture meteorological characteristics and generate more accurate and informative reports across diverse temporal contexts.

### D.3 NWP-AUGMENTED TRAINING

Weather forecasting pipelines are not limited to using only the initial state as visual input. In fact, NWP forecasting results can also be incorporated to provide additional visual information. Specifically, we retrieve the HRES forecasting outputs from WeatherBench2, which include the 00 and 12 UTC initializations of HRES along with their corresponding 6-hourly forecasts. We select six single-level variables: 10 m u-component of wind, 10 m v-component of wind, 2 m temperature, mean sea level pressure, surface pressure, and 6-hour total precipitation. To reduce computational complexity caused by the large number of images, we compute the mean value of each prediction over 24 hours, thereby obtaining daily prediction variables spanning from 1 to 5 days. During training, we replace the visual data in **WFInstruct** with variables from different prediction horizons, ranging from only 1-day predictions to the full 1–5-day forecasts, in order to assess whether incorporating longer-range forecast information can further enhance performance. The results are reported in Table 4. We can observe that as the time horizon of the visual input increases, the model's performance improves accordingly, demonstrating that incorporating NWP prediction results can further enhance performance. A marginal effect is observed as the time horizon extends from 3 days to 4 days. However, as the horizon increases from 4 to 5 days, the performance in some cities declines. This decrease can be attributed to the fact that the extensive visual input introduces unnecessary noise. Not all cities experience an increase in performance as the time horizon expands. For OKX, the best performance is achieved with only 1-day visual data input.

Table 4: The impact of different prediciton horizon on training

| Horizon | HK | BOX | LOX | LWX | MTR | OKX | PBZ | PQR | SEW | VEF | Average |
|---------|-----|-----|-----|-----|-----|-----|-----|-----|-----|-----|---------|
| | | | | | *Correctness* | | | | | | |
| 1Day | 3.31 | 2.41 | 2.40 | 3.06 | 2.67 | 3.45 | 2.89 | 2.68 | 2.91 | 2.44 | 2.82 |
| 1-2Day | 3.35 | 2.66 | 2.50 | 3.12 | 2.81 | 3.45 | 2.86 | 2.78 | 2.96 | 2.44 | 2.89 |
| 1-3Day | 3.60 | 2.84 | 2.81 | 3.17 | 2.91 | 3.44 | 3.13 | 2.90 | 2.98 | 2.83 | 3.06 |
| 1-4Day | 3.64 | 2.83 | 2.87 | 3.16 | 3.08 | 3.45 | 3.14 | 2.97 | 3.02 | 2.85 | 3.10 |
| 1-5Day | 3.54 | 2.78 | 2.84 | 3.09 | 3.03 | 3.40 | 3.01 | 3.07 | 3.24 | 2.84 | 3.07 |

## D.4 ABLATION STUDY OF RFT

To further verify that the performance gain of RFT does not simply result from an increased training volume (i.e., more epochs), but rather from the incorporation of diverse reject-sampled data, we conduct an experiment on **WFInstruct** with five training epochs. The results are reported in Table 5. We observe that the performance of training for five epochs is similar to that of training for a single epoch, and both are significantly worse than RFT. This highlights that the key advantage of RFT lies in leveraging diverse reject-sampled data to enhance model generalization, rather than in longer or larger-scale training.

Table 5: Performances of **SynopticMind** and its variants on **WFInstruct** test set.

| Algorithms | HK | BOX | LOX | LWX | MTR | OKX | PBZ | PQR | SEW | VEF | Average |
|------------|-----|-----|-----|-----|-----|-----|-----|-----|-----|-----|---------|
| | | | | | *Correctness* | | | | | | |
| **SynopticMind** | 3.66 | 2.79 | 2.58 | 3.05 | 2.81 | 3.28 | 2.85 | 2.73 | 2.93 | 2.63 | 2.93 |
| **SynopticMind**(5 epoch) | 3.68 | 2.80 | 2.63 | 3.08 | 2.85 | 3.38 | 2.91 | 2.82 | 2.91 | 2.60 | 2.96 |
| **SynopticMind-RFT** | 3.70 | 2.90 | 3.14 | 3.31 | 3.03 | 3.54 | 3.02 | 3.16 | 3.43 | 2.90 | 3.21 |

## E  MORE DATASET DETAILS

### E.1 DETAILS OF TEXTUAL DATA

We perform weather forecast report in 10 cities, including Hong Kong (HK), Boston (BOX), Los Angles (LOX), Washington DC (LWX), San Francisco (MTR), New York (OKX), Pittsburgh (PBZ), Portland (PQR), Seattle (SEW) and Las Vegas (VEF). For the daily reports in Hong Kong, we obtain report data from the Hong Kong Open Data Platform[3]. The textual reports focus on short-range same-day weather forecasts. For the other nine cities, we retrieve the synopsis data from the Area Forecast Discussion text products provided by the Iowa Environmental Mesonet[4], which archive past short- and medium-range weather forecast synopsis.

### E.2 ALIGNMENT OF THE ERA5 AND REPORT

We first convert the UTC timestamps in the ERA5 data to the local time for each city. Since the temporal resolution is 6 hours, we select the report issued within the following 3 hours as the corresponding report for the ERA5 visual input. As multiple reports may be issued throughout a single day, we use the one released in the early morning as the representative report for that day. The example reports are shown in Table6.

---

[2]* API currently can process a maximum of 16 images, limiting our test to the 0-shot setting.

[3]https://data.gov.hk/sc-data/dataset/hk-hko-rss-local-weather-forecast

[4]https://mesonet.agron.iastate.edu/wx/afos/

### E.3 DETAILS OF **WFINSTRUCT-PRESSURE** AND **WFINSTRUCT-GLOBAL**

**WFInstruct-Pressure**   To explore the impact of incorporating pressure-level variables, we construct an extended dataset, called **WFInstruct-Pressure**. Specifically, we supplement the regional single-level data with pressure-level variables including geopotential, specific humidity, temperature, and the u- and v-components of wind at four standard pressure levels: 200 hPa, 500 hPa, 700 hPa, and 850 hPa. The detailed information is shown in Appendix Table 9. This enriched dataset allows us to assess whether vertical atmospheric structure contributes to improved model performance.

**WFInstruct-Global**   To evaluate the influence of broader spatial context, we create another dataset, **WFInstruct-Global**, by incorporating large-scale geographical information. For the Hong Kong region, we include additional data from the surrounding regions of China. For regions in the United States, we add single-level data covering the entire CONUS area. This setup allows us to examine how the incorporation of a wider regional view affects the forecasting ability of the model.

## F   PROMPT

The prompts in **WFInstruct** are shown in Table10. The zero-shot and few-shot prompts for closed-source model are shown in Table11 and Table12. The prompt for evaluation are shown in Table13 and Table14.

## G   MORE CASES

More cases are shown in Figure9 to Figure17

Table 6: The example report for each city

| Area & Time | Report Example |
|---|---|
| HK(2022-01-01T08) | Mainly fine and dry. The maximum temperature will be around 21 degrees. Moderate to fresh east to northeasterly winds. |
| BOX(2022-01-02T07) | Mostly cloudy conditions today with continued above normal temperatures, with decreasing clouds and falling temperatures to more seasonable levels Sunday night. Cold and blustery on Monday with the potential for minor accumulating snowfall along the south coast into Cape Cod and the Islands. Continued cold conditions on Tuesday and then temperatures moderate to above normal Wednesday, although a period of showers are likely. Trending colder late in the week with rain or snow possible. |
| LOX(2022-01-01T10) | Breezy north to northeast winds expected this weekend under clear skies. High temperatures will warm slightly but remain several degrees below normal. A mostly dry period expected for the next couple weeks except for possibly some very light precip along the Central Coast Friday. |
| LWX(2022-01-01T07) | A warm front will lift north into the area late today and tonight as one wave of low pressure moves through. This front will then slide back south early Sunday while a stronger cold front will sweep across the area late Sunday. Another wave of low pressure will pass southeast of the area late Sunday night and Monday. High pressure will return briefly Tuesday. Another system with potential winter implications arrives by Thursday across the mountains. |
| MTR(2022-01-01T10) | Cool and dry conditions will prevail for the first part of the new year, with unseasonably cold overnight and early morning temperatures through this weekend. The chance of more beneficial rain returns for the work week, with temperatures returning to more normal ranges. |
| OKX(2022-01-01T07) | A frontal system impacts the forecast area this weekend, with a cold front moving through late tonight into Sunday morning with an area of low pressure. High pressure then builds in from the west Monday through Tuesday. One frontal system passes well north of the area Wednesday into Thursday while another crosses the region Thursday night into Friday. |
| PBZ(2022-01-01T07) | A wet start to the New Year, but temperatures will be mild. Much colder on Sunday. |
| PQR(2022-01-01T10) | Chilly and dry today as high pressure provides a break in the weather. Rain returns on Sunday, with wintry weather in the central Columbia Gorge and Hood River, as well as snow in the mountains. Rain, with more seasonable temperatures expected next week. Snow will continue to pile up in the Cascades. |
| SEW(2022-01-02T16) | A cold front will move inland tonight with lowland rain and heavy mountain snow. The next system moves into Oregon on Tuesday with Fraser river outflow winds developing over Whatcom county. A system moving up from the southwest Wednesday night into Thursday will bring more lowland rain, chance of snow Everett north and heavy snow in the mountains. |
| VEF(2022-01-01T10) | Dry but chilly weather is in store for the area to begin the new year with temperatures around 10 degrees below normal over the weekend. Gusty north winds today will make conditions feel much colder but lighter winds on Sunday should make it more tolerable. High pressure will lead to a warming trend next week with above normal temperatures expected by Wednesday. |

Table 7: Variable categories in variable analysis experiment

| group | variable |
|-------|----------|
| WND | 10m u component of wind, 10m v component of wind |
| TMP | 2m temperature, sea surface temperature |
| WTR | total precipitation 6hr, total column water vapour, total column water |
| PRS | surface pressure, mean sea level pressure |
| CLD | total cloud cover |
| SNW | snow depth |

Table 8: The definition of single level variable.

| Variable | Definition |
|----------|------------|
| land sea mask | The proportion of land, as opposed to ocean or inland waters |
| 10m u component of wind | The eastward component of the 10m wind. It is the horizontal speed of air moving towards the east, at a height of ten metres above the surface of the Earth. |
| 10m v component of wind | The northward component of the 10m wind. It is the horizontal speed of air moving towards the north, at a height of ten metres above the surface of the Earth. |
| 2m temperature | The temperature of air at 2m above the surface of land, sea or in-land waters. |
| mean sea level pressure | The pressure of the atmosphere adjusted to the height of mean sea level. |
| sea surface temperature | The temperature of sea water near the surface. |
| snow depth | The depth of snow from the snow-covered area. |
| surface pressure | The pressure of the atmosphere on the surface of land, sea and in-land water. |
| total cloud cover | The proportion of a grid box area covered by cloud. |
| total precipitation 6hr | The total precipitation over the past 6 hours. |
| total column water vapour | The total amount of water vapour in a vertical column of the atmosphere, from the surface to the top of the atmosphere. |
| total column water | The total amount of liquid water in a vertical column of the atmosphere. |

Table 9: The definition of pressure level variable, which is available on multiple levels through the atmosphere.

| Variable | Definition |
|---|---|
| geopotential | This parameter is the gravitational potential energy of a unit mass, at a particular location, relative to mean sea level. |
| specific humidity | This parameter is the mass of water vapour per kilogram of moist air. The total mass of moist air is the sum of the dry air, water vapour, cloud liquid, cloud ice, rain and falling snow. |
| temperature | This parameter is the temperature in the atmosphere. |
| u component of wind | This parameter is the eastward component of the wind. It is the horizontal speed of air moving towards the east, in metres per second. |
| v component of wind | This parameter is the northward component of the wind. It is the horizontal speed of air moving towards the north, in metres per second. |

Table 10: Prompt in **WFInstruct**

As an AI assistant with expertise in weather forecasting, you are equipped to interpret comprehensive figures that illustrate various weather variables crucial for understanding the latest weather conditions across the [City]. Your responsibility as a weather forecaster is to provide accurate and timely insights into weather conditions. The following figures represent weather conditions at [Year-Month-Day], [Hour], [Weekday], and each figure contains one weather parameter.<image><image><image><image><image><image><image><image> <image><image><image><image>. Generate a concise natural language weather forecast report based on the figures.

Table 11: Zero shot prompt for baselines

| System Prompt | As an AI assistant with expertise in severe weather analysis and forecasting, you are equipped to interpret comprehensive figures that illustrate various weather variables crucial for understanding the latest weather conditions across [City]. Your responsibility as a weather forecaster is to produce a general weather forecast for the future using the current weather condition images provided. |
|---|---|
| User Prompt | <Thought process> Use the following clues to generate a concise natural language weather forecast report based on the figures.: 1. Go through each depicted weather field within the figures in <Parameters> for weather analysis one by one and consider whether each weather analysis field shown signs of potential for future weather. 2. Focus on general weather forecast descriptions instead of overly detailed reports. Output the general weather forecast report directly. </Thought process> <Question>Please analyze the following figures and directly generate a concise overall weather forecast based on the current observed patterns. You do not need to describe each existing individual image; instead, directly provide a unified forecast report.</Question> [Question Prompt] |
| [Question Prompt] | <Question><Parameters> The following 12 figures represent weather conditions at [Year-Month-Day, Hour, Weekday] and each figure contains multiple weather parameters, the most important variable in each figure is provided as follows:[Image Description] </Parameters> </Question> |
| [Image Description] | [Definition] <Image>, [Definition] <Image>, [Definition] <Image>, [Definition] <Image>, [Definition] <Image>, [Definition] <Image>, [Definition] <Image>, [Definition] <Image>, [Definition] <Image>, [Definition] <Image>, [Definition] <Image>, [Definition] <Image> |

Table 12: Few-shot prompt for baselines

| | |
|---|---|
| System Prompt | As an AI assistant with expertise in severe weather analysis and forecasting, you are equipped to interpret comprehensive figures that illustrate various weather variables crucial for understanding the latest weather conditions across [City]. Your responsibility as a weather forecaster is to produce a general weather forecast for the future using the current weather condition images provided. |
| User Prompt | <Thought process> Use the following clues to generate a concise natural language weather forecast report based on the figures.: 1. Go through each depicted weather field within the figures in <Parameters> for weather analysis one by one and consider whether each weather analysis field shown signs of potential for future weather. 2. Focus on general weather forecast descriptions instead of overly detailed reports. Output the general weather forecast report directly. </Thought process> <Question>Please analyze the following figures and directly generate a concise overall weather forecast based on the current observed patterns. You do not need to describe each existing individual image; instead, directly provide a unified forecast report.</Question> <Examples> Below are a few examples of weather analysis to help understand how the region and type of concern relate to different weather conditions: [Example Prompt](Repeat N times for few-shot)</Examples>[Question Prompt] |
| [Example Prompt] | <Example [id]><Parameters> The following 12 figures represent weather conditions at [Year-Month-Day, Hour, Weekday] and each figure contains multiple weather parameters, the most important variable in each figure is provided as follows: [Image Description][Example Answers]</Parameters></Example [id]> |
| [Image Description] | [Definition] <Image>, [Definition] <Image>, [Definition] <Image>, [Definition] <Image>, [Definition] <Image>, [Definition] <Image>, [Definition] <Image>, [Definition] <Image>, [Definition] <Image>, [Definition] <Image>, [Definition] <Image>, [Definition] <Image> |
| [Question Prompt] | <Question><Parameters> The following 12 figures represent weather conditions at [Year-Month-Day, Hour, Weekday] and each figure contains multiple weather parameters, the most important variable in each figure is provided as follows:[Image Description] </Parameters> </Question> |

Table 13: Prompts for correctness evaluation

| | |
|---|---|
| System prompt | You are an intelligent evaluator designed to assess the factual consistency between a generated weather forecast and the ground truth forecast. Your task is to carefully compare the predicted forecast text with the correct (ground truth) forecast and determine how factually aligned they are. INSTRUCTIONS: Focus on the factual consistency between the predicted answer and the correct answer. The predicted answer should not contain any misinterpretations or misinformation. Minor paraphrasing or use of synonyms is acceptable as long as the core facts match. Your evaluation must be based only on the factual content, not language fluency or completeness. |
| User Prompt | Please evaluate the following ground truth and prediction answer pair: Correct Answer: [...] Predicted Answer:[...] Provide your evaluation only as a factual accuracy score where the factual accuracy score is an integer value between 0 and 5, with 5 indicating the highest level of factual consistency. Please generate the response in the form of a Python dictionary string with keys 'score', where its value is the factual accuracy score in INTEGER, not STRING. DO NOT PROVIDE ANY OTHER OUTPUT TEXT OR EXPLANATION. Only provide the Python dictionary string. For example, your response should look like this: {'score': 4.8}. |

Table 14: Prompts for detailedness evaluation

| System prompt | You are an intelligent evaluator designed to assess the detail orientation of predicted weather forecasts compared to ground truth forecasts. Your task is to determine how complete and specific the predicted forecast is relative to the ground truth. INSTRUCTIONS: Evaluate whether the predicted forecast covers all key weather elements mentioned in the ground truth. Assess the specificity of the prediction: it should include detailed and context-relevant descriptions rather than vague or generic phrases. Minor paraphrasing or use of synonyms is acceptable as long as the core facts match. |
|---|---|
| User Prompt | Please evaluate the following ground truth and prediction answer pair: Correct Answer: [...] Predicted Answer:[...] Provide your evaluation only as a factual accuracy score where the factual accuracy score is an integer value between 0 and 5, with 5 indicating the highest level of factual consistency. Please generate the response in the form of a Python dictionary string with keys 'score', where its value is the factual accuracy score in INTEGER, not STRING. DO NOT PROVIDE ANY OTHER OUTPUT TEXT OR EXPLANATION. Only provide the Python dictionary string. For example, your response should look like this: {'score': 4.8}. |

**Question:**
As an AI assistant with expertise in weather forecasting, you are equipped to interpret a comprehensive figures that illustrating various weather variables crucial for understanding the latest weather conditions across the Hong Kong.
Your responsibility as a weather forecaster is to provide accurate and timely insights into weather conditions.
The following figures represent weather conditions at 2022-03-10, 08 hour, Thursday and each figure contains one weather parameters.\n<image>\n\n<image>\n\n<image>\n\n<image>\n\n<image>\n\n<image>\n\n<image>\n\n<image>\n\n<image>\n\n<image>\n\n<image>
Generate a concise natural language weather forecast report based on the figures.

**Visual Input:**

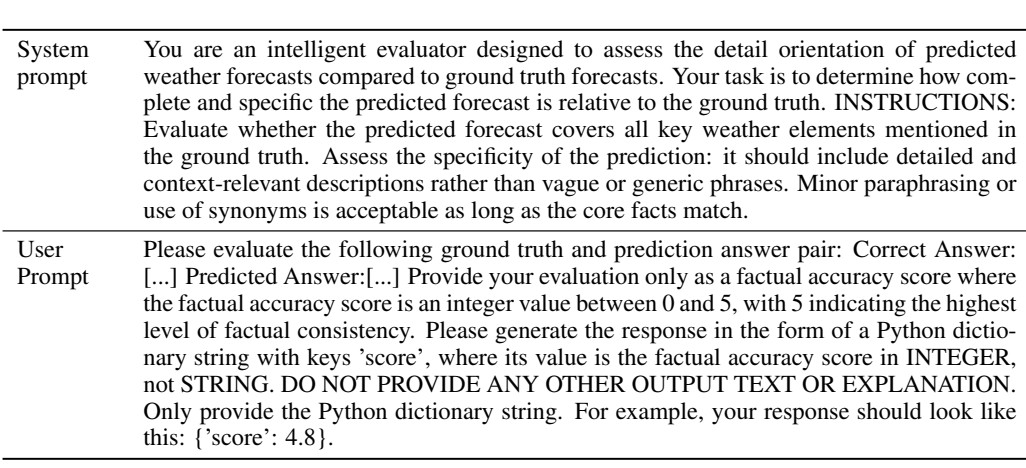

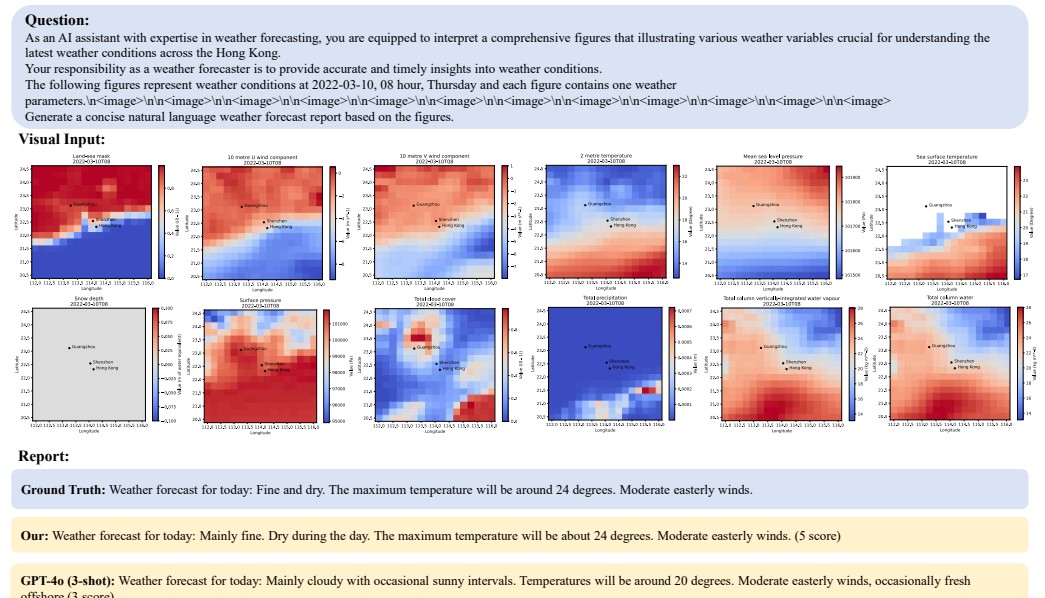

**Report:**

**Ground Truth:** Weather forecast for today: Fine and dry. The maximum temperature will be around 24 degrees. Moderate easterly winds.

**Our:** Weather forecast for today: Mainly fine. Dry during the day. The maximum temperature will be about 24 degrees. Moderate easterly winds. (5 score)

**GPT-4o (3-shot):** Weather forecast for today: Mainly cloudy with occasional sunny intervals. Temperatures will be around 20 degrees. Moderate easterly winds, occasionally fresh offshore.(3 score)

Figure 9: A typical case for Hong Kong

**Question:**
As an AI assistant with expertise in weather forecasting, you are equipped to interpret a comprehensive figures that illustrating various weather variables crucial for understanding the latest weather conditions across the Boston.
Your responsibility as a weather forecaster is to provide accurate and timely insights into weather conditions.
The following figures represent weather conditions at 2022-06-20, 08 hour, Thursday and each figure contains one weather
parameters.\n\n<image>\n\n<image>\n\n<image>\n\n<image>\n\n<image>\n\n<image>\n\n<image>\n\n<image>\n\n<image>\n\n<image>\n\n<image>
Generate a concise natural language weather forecast report based on the figures.

**Visual Input Example:**

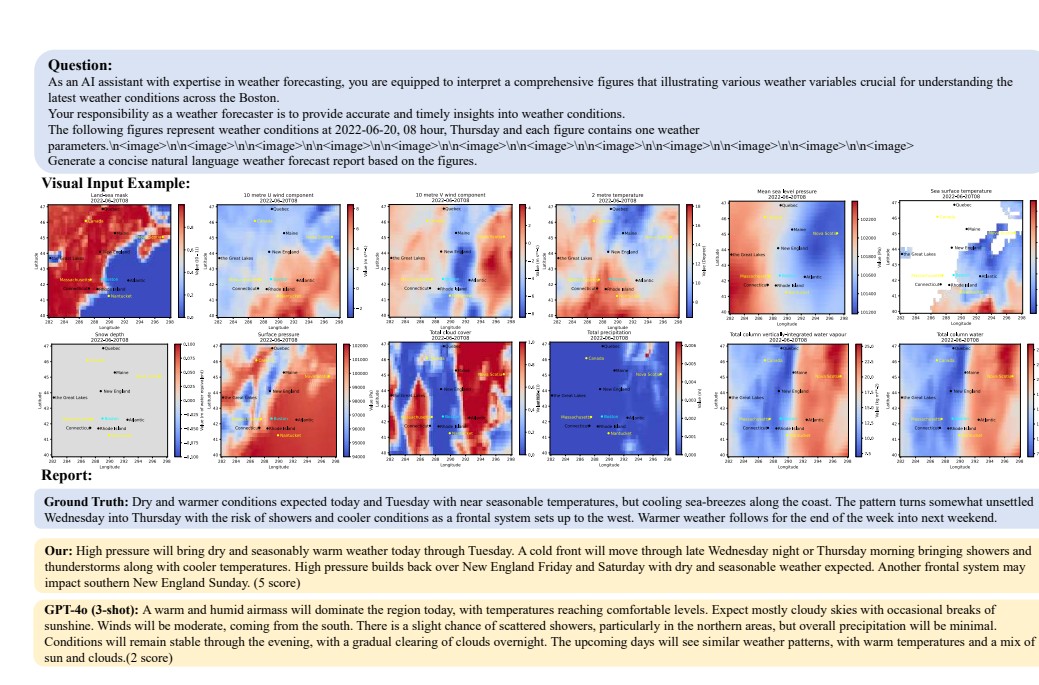

**Report:**

**Ground Truth:** Dry and warmer conditions expected today and Tuesday with near seasonable temperatures, but cooling sea-breezes along the coast. The pattern turns somewhat unsettled Wednesday into Thursday with the risk of showers and cooler conditions as a frontal system sets up to the west. Warmer weather follows for the end of the week into next weekend.

**Our:** High pressure will bring dry and seasonably warm weather today through Tuesday. A cold front will move through late Wednesday night or Thursday morning bringing showers and thunderstorms along with cooler temperatures. High pressure builds back over New England Friday and Saturday with dry and seasonable weather expected. Another frontal system may impact southern New England Sunday. (5 score)

**GPT-4o (3-shot):** A warm and humid airmass will dominate the region today, with temperatures reaching comfortable levels. Expect mostly cloudy skies with occasional breaks of sunshine. Winds will be moderate, coming from the south. There is a slight chance of scattered showers, particularly in the northern areas, but overall precipitation will be minimal. Conditions will remain stable through the evening, with a gradual clearing of clouds overnight. The upcoming days will see similar weather patterns, with warm temperatures and a mix of sun and clouds.(2 score)

Figure 10: A typical case for Boston

**Question:**
As an AI assistant with expertise in weather forecasting, you are equipped to interpret a comprehensive figures that illustrating various weather variables crucial for understanding the latest weather conditions across the Los Angeles.
Your responsibility as a weather forecaster is to provide accurate and timely insights into weather conditions.
The following figures represent weather conditions at 2022-08-16, 05 hour, Tuesday and each figure contains one weather
parameters.\n\n<image>\n\n<image>\n\n<image>\n\n<image>\n\n<image>\n\n<image>\n\n<image>\n\n<image>\n\n<image>\n\n<image>\n\n<image>
Generate a concise natural language weather forecast report based on the figures.

**Visual Input Example:**

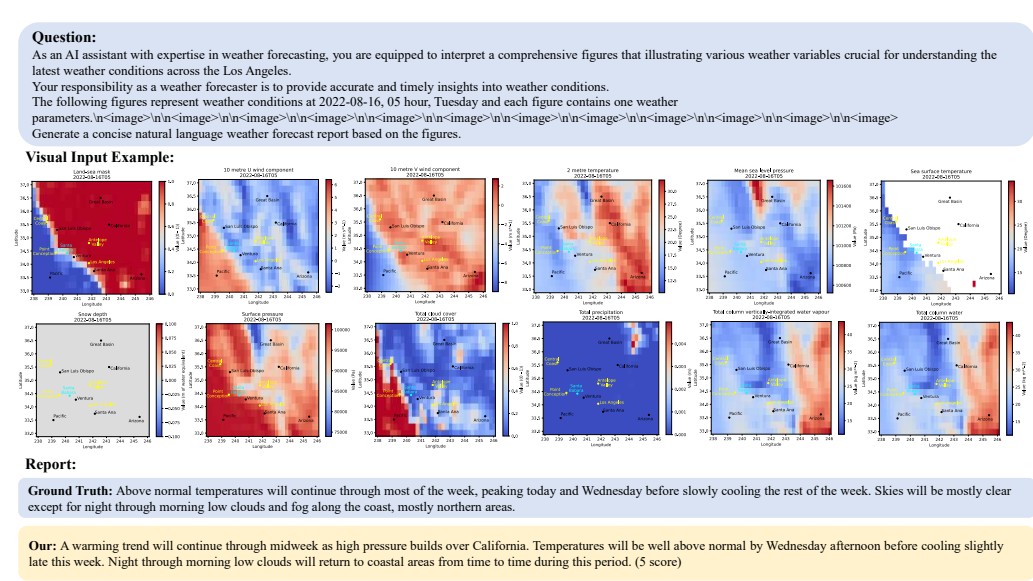

**Report:**

**Ground Truth:** Above normal temperatures will continue through most of the week, peaking today and Wednesday before slowly cooling the rest of the week. Skies will be mostly clear except for night through morning low clouds and fog along the coast, mostly northern areas.

**Our:** A warming trend will continue through midweek as high pressure builds over California. Temperatures will be well above normal by Wednesday afternoon before cooling slightly late this week. Night through morning low clouds will return to coastal areas from time to time during this period. (5 score)

**GPT-4o (3-shot):** Hot and dry conditions will persist across the region, with temperatures peaking in the interior areas. Coastal areas may experience some relief with cooler sea breezes. Minimal cloud cover is expected, and there is no significant precipitation forecasted. Winds will be generally light, with occasional gusts in the valleys.(3 score)

Figure 11: A typical case for Los Angeles

**Question:**
As an AI assistant with expertise in weather forecasting, you are equipped to interpret a comprehensive figures that illustrating various weather variables crucial for understanding the latest weather conditions across the San Francisco.
Your responsibility as a weather forecaster is to provide accurate and timely insights into weather conditions.
The following figures represent weather conditions at 2022-10-17, 05 hour, Monday and each figure contains one weather parameters.\n\n<image>\n\n<image>\n\n<image>\n\n<image>\n\n<image>\n\n<image>\n\n<image>\n\n<image>\n\n<image>\n\n<image>\n\n<image>
Generate a concise natural language weather forecast report based on the figures.

**Visual Input Example:**

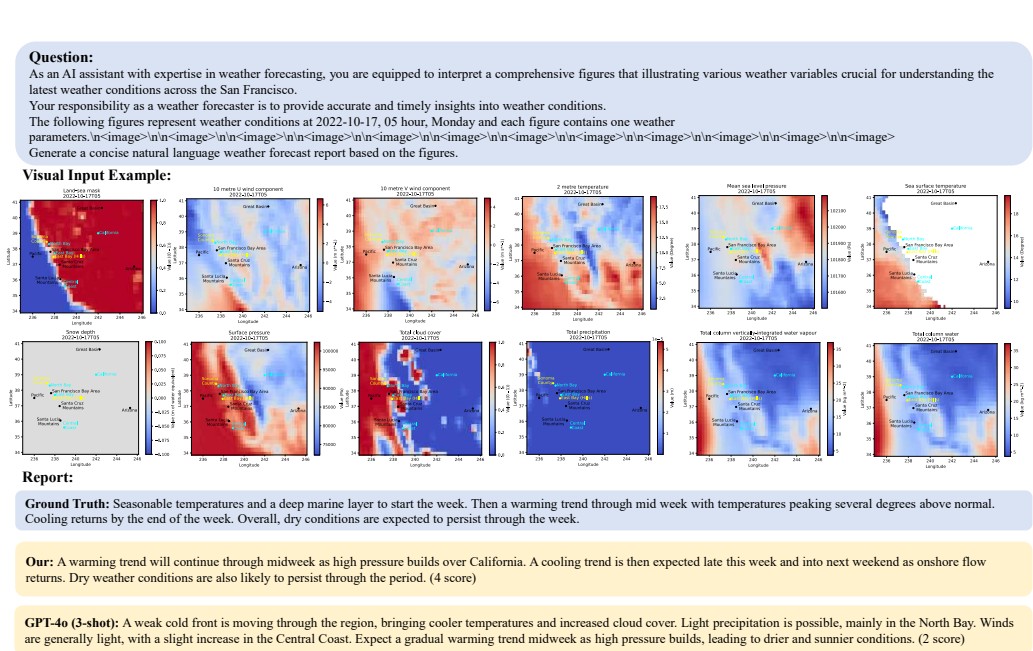

**Report:**

**Ground Truth:** Seasonable temperatures and a deep marine layer to start the week. Then a warming trend through mid week with temperatures peaking several degrees above normal. Cooling returns by the end of the week. Overall, dry conditions are expected to persist through the week.

**Our:** A warming trend will continue through midweek as high pressure builds over California. A cooling trend is then expected late this week and into next weekend as onshore flow returns. Dry weather conditions are also likely to persist through the period. (4 score)

**GPT-4o (3-shot):** A weak cold front is moving through the region, bringing cooler temperatures and increased cloud cover. Light precipitation is possible, mainly in the North Bay. Winds are generally light, with a slight increase in the Central Coast. Expect a gradual warming trend midweek as high pressure builds, leading to drier and sunnier conditions. (2 score)

Figure 12: A typical case for San Francisco

**Question:**
As an AI assistant with expertise in weather forecasting, you are equipped to interpret a comprehensive figures that illustrating various weather variables crucial for understanding the latest weather conditions across the Pittsburgh.
Your responsibility as a weather forecaster is to provide accurate and timely insights into weather conditions.
The following figures represent weather conditions at 2022-05-30, 08 hour, Monday and each figure contains one weather parameters.\n\n<image>\n\n<image>\n\n<image>\n\n<image>\n\n<image>\n\n<image>\n\n<image>\n\n<image>\n\n<image>\n\n<image>\n\n<image>
Generate a concise natural language weather forecast report based on the figures.

**Visual Input Example:**

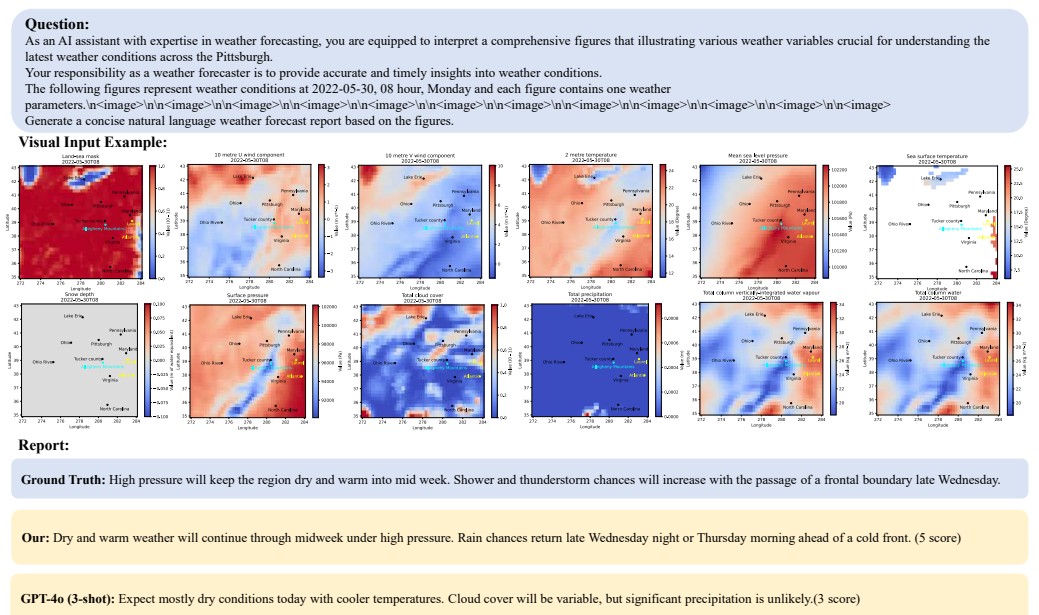

**Report:**

**Ground Truth:** High pressure will keep the region dry and warm into mid week. Shower and thunderstorm chances will increase with the passage of a frontal boundary late Wednesday.

**Our:** Dry and warm weather will continue through midweek under high pressure. Rain chances return late Wednesday night or Thursday morning ahead of a cold front. (5 score)

**GPT-4o (3-shot):** Expect mostly dry conditions today with cooler temperatures. Cloud cover will be variable, but significant precipitation is unlikely.(3 score)

Figure 13: A typical case for Pittsburgh

**Question:**
As an AI assistant with expertise in weather forecasting, you are equipped to interpret a comprehensive figures that illustrating various weather variables crucial for understanding the latest weather conditions across the Portland.
Your responsibility as a weather forecaster is to provide accurate and timely insights into weather conditions.
The following figures represent weather conditions at 2022-01-05, 10 hour, Wednesday and each figure contains one weather
parameters.\n\n<image>\n\n<image>\n\n<image>\n\n<image>\n\n<image>\n\n<image>\n\n<image>\n\n<image>\n\n<image>\n\n<image>
Generate a concise natural language weather forecast report based on the figures.

**Visual Input Example:**

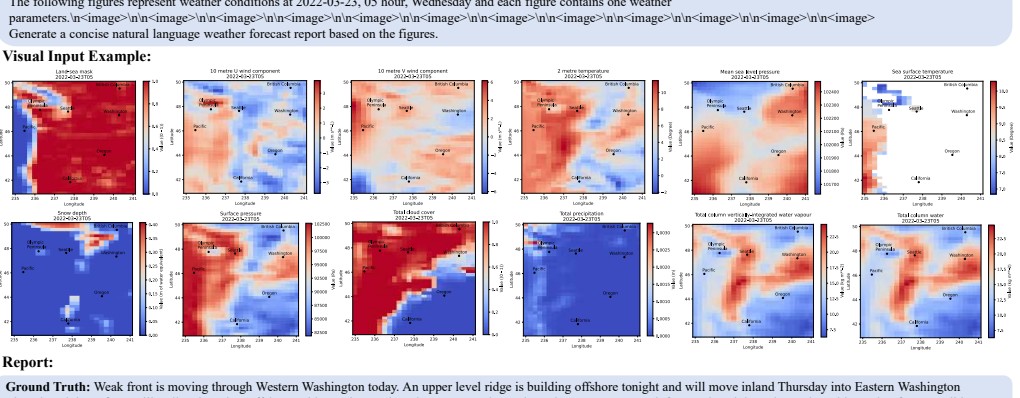

**Report:**

**Ground Truth:** A warm front will lift slowly northward across the forecast area today and tonight, bringing rising snow levels and another round of rain.\t Cool air trapped in the Hood River Valley will ensure snow and freezing rain later tonight into Thursday morning.   Wet and mild on Thu. Then, another cold front will push onshore early Fri, with cooler air. High pressure will bring dry but cool weather for Sat and Sun, but rain returns next week.

**Our:** A weak front will move through the region today bringing light rain or snow showers mainly north of Salem. A stronger system will bring more widespread precipitation Thursday night and Friday morning before high pressure builds over the weekend. Another frontal system may impact the area early next week. (4 score)

**GPT-4o (3-shot):** A high-pressure system is dominating the region, leading to dry and cold conditions. Winds are generally light, with a slight eastward component. Temperatures are low, especially inland, with some areas experiencing below-freezing conditions. Snow depth is minimal, but some accumulation is present in higher elevations. Cloud cover is sparse, contributing to the cold temperatures. Precipitation is low, but some isolated areas may experience light snow or rain. The weather is expected to remain stable, with no significant systems approaching in the immediate future.(1 score)

Figure 14: A typical case for Portland

**Question:**
As an AI assistant with expertise in weather forecasting, you are equipped to interpret a comprehensive figures that illustrating various weather variables crucial for understanding the latest weather conditions across the Seattle.
Your responsibility as a weather forecaster is to provide accurate and timely insights into weather conditions.
The following figures represent weather conditions at 2022-03-23, 05 hour, Wednesday and each figure contains one weather
parameters.\n\n<image>\n\n<image>\n\n<image>\n\n<image>\n\n<image>\n\n<image>\n\n<image>\n\n<image>\n\n<image>\n\n<image>
Generate a concise natural language weather forecast report based on the figures.

**Visual Input Example:**

**Report:**

**Ground Truth:** Weak front is moving through Western Washington today. An upper level ridge is building offshore tonight and will move inland Thursday into Eastern Washington Thursday night. A front will stall and weaken offshore Friday and move into the area Saturday and Sunday. An upper trough for Sunday night and Monday with another front possible on Tuesday.

**Our:** A weak upper level ridge will build over Western Washington today bringing dry weather through Thursday morning. A frontal system will move inland late Friday night or Saturday bringing rain and mountain snow back to the area. Drier weather returns Sunday before another front arrives early next week. (4 score)

**GPT-4o (3-shot):** A weak frontal system will bring light rain to the Olympic Peninsula today, with mostly cloudy skies across the region. Winds will be light and variable. Temperatures will remain mild, with a slight cooling trend expected. High pressure will build later in the week, leading to drier conditions and clearing skies. (2 score)

Figure 15: A typical case for Seattle

**Question:**
As an AI assistant with expertise in weather forecasting, you are equipped to interpret a comprehensive figures that illustrating various weather variables crucial for understanding the latest weather conditions across the Las Vegas.
Your responsibility as a weather forecaster is to provide accurate and timely insights into weather conditions.
The following figures represent weather conditions at 2022-01-29, 10 hour, Saturday and each figure contains one weather parameters.\n<image>\n\n<image>\n\n<image>\n\n<image>\n\n<image>\n\n<image>\n\n<image>\n\n<image>\n\n<image>\n\n<image>\n\n<image>
Generate a concise natural language weather forecast report based on the figures.

**Visual Input Example:**

**Report:**

**Ground Truth:** Widespread mid and high clouds will gradually clear tonight with mostly sunny skies in store for Sunday. Pleasant conditions are expected through early next week with a cold front bringing cooler and blustery weather by midweek.

**Our:** Dry and mild conditions will continue through Sunday before another storm system brings increasing winds Monday along with cooler temperatures Tuesday and Wednesday. Dry conditions return late next week. (4 score)

**GPT-4o (3-shot):** Expect mostly dry and mild conditions today with light winds across the region. Cloud cover will be moderate, but no significant precipitation is anticipated. Temperatures will remain slightly below normal. A high-pressure system is maintaining stable weather, but changes are expected early next week with a potential for cooler temperatures and increased cloudiness.(3 score)

Figure 16: A typical case for Las Vegas

**Question:**
As an AI assistant with expertise in weather forecasting, you are equipped to interpret a comprehensive figures that illustrating various weather variables crucial for understanding the latest weather conditions across the Washington DC.
Your responsibility as a weather forecaster is to provide accurate and timely insights into weather conditions.
The following figures represent weather conditions at 2022-03-31, 08 hour, Thursday and each figure contains one weather parameters.\n<image>\n\n<image>\n\n<image>\n\n<image>\n\n<image>\n\n<image>\n\n<image>\n\n<image>\n\n<image>\n\n<image>\n\n<image>
Generate a concise natural language weather forecast report based on the figures.

**Visual Input Example:**

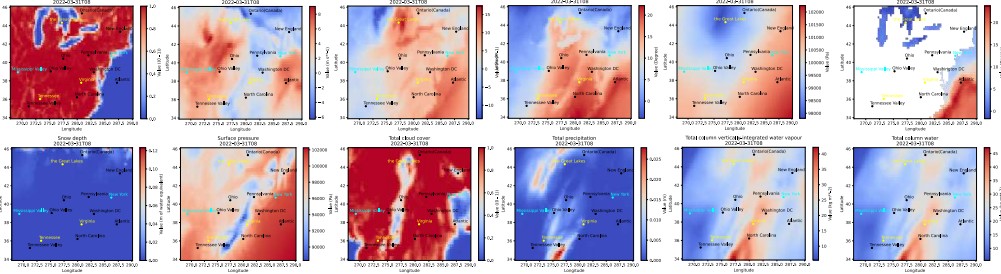

**Report:**

**Ground Truth:** A strong cold front will approach the area later today, then cross the area tonight. High pressure will build in Friday into Saturday. A cold front and wave of low pressure are expected to move over the area Saturday night into Sunday. High pressure is expected to make another brief return early next week before unsettled weather returns by the middle of next week.

**Our:** A cold front will cross the region this afternoon followed by high pressure building over the area Friday through Saturday night. Another low pressure system may impact the area Sunday before high pressure returns early next week. (5 score)

**GPT-4o (3-shot):** A cold front will move through the region today, bringing cooler temperatures and increased cloud cover. High pressure will build in from the west, leading to drier conditions by tomorrow. Expect some scattered showers, particularly in the morning, with clearing skies later in the day. Temperatures will remain mild, with a gradual warming trend into the weekend. (2 score)

Figure 17: A typical case for Washington DC

