# OpenReview forum: "SynopticMind: An Instruction Tuning MLLM For Weather  Forecasting Report Generation"
_ICLR.cc/2026/Conference — ICLR 2026 Conference Withdrawn Submission_

### Official Review · Reviewer_YHTv · 2025-10-23

**Soundness:** 1
**Presentation:** 3
**Contribution:** 2
**Rating:** 2
**Confidence:** 4

**Summary:**

The paper introduces a multimodal task and model for generating human-style weather forecast reports from ERA5 heatmaps. The pipeline trains a Qwen-VL variant via SFT to “RFT” (mining multiple generations) to DPO, and evaluates with an LLM-as-judge on two rubric dimensions. Results show improvements over several baselines on the authors’ dataset.

**Strengths:**

1. The authors tackle the important problem of generating weather reports from ERA5 data.
2. The paper is well-presented and relatively easy to follow along.
3. The authors include code, the complete dataset, and other artifacts are included as part of the submission, aiding reproducibility and comprising of a substantial contribution.

**Weaknesses:**

1. The same class of LLMs is used to (a) score intermediate generations during RFT/DPO and (b) score final outputs at evaluation time. This design creates a closed loop where the model can learn judge-specific artifacts rather than meteorological correctness, heightening the risk of reward hacking.

2. The 0–5 “factual accuracy” prompt lacks a rubric defining what each score means and provides no protocol for repeated sampling or inter-rater checks. Prior work [1, 2] shows LLM evaluators can be inconsistent and biased, which undermines conclusions drawn from a single judge with a vague scale. A stronger setup would (i) define clear rubrics that define the value for each ordinal of the produced score (ii) diversify (or swap) the judge model at evaluation, (iii) include human forecaster side-by-side, or at the very least, check the alignment between human expert scores and LLM scores.

3. RFT amplifies samples produced by the same model family and selected by the same judge, encouraging alignment to judge-preferred phrasing rather than ground-truth meteorology. A more principled alternative is to paraphrase the gold reports with a text LLM (with factuality checks) to increase lexical diversity while preserving facts. If self-augmentation is retained, the paper should quantify factual drift against ERA5 signals and compare against gold-paraphrase augmentation.


### References

[1] Lee, Noah, Jiwoo Hong, and James Thorne. "Evaluating the consistency of llm evaluators." arXiv preprint arXiv:2412.00543 (2024).
[2] Stureborg, Rickard, Dimitris Alikaniotis, and Yoshi Suhara. "Large language models are inconsistent and biased evaluators." arXiv preprint arXiv:2405.01724 (2024).

**Questions:**

### Questions (in addition to points raised in the Weaknesses section)

1. You set a threshold of 5 for HK but 4 for other cities when selecting RFT samples. Why is HK strictly filtered while others are looser?
2. Which fields from ERA5 were given to the model? How was the context provided, i.e. as a single image or multiple images? This should be stated clearly in the main paper.



### Suggestions

1. Improve the grading rubric and compare against human expert evaluators (if possible). Report more diverse metrics that focus on factuality.
2. Line 85: typo ('extensove'). Line 128: Direct Preferred Optimization should be Direct Preference Optimization.
3. Consider writing a more descriptive caption for Figure 2.

---

### Official Review · Reviewer_HBEr · 2025-10-30

**Soundness:** 2
**Presentation:** 3
**Contribution:** 2
**Rating:** 2
**Confidence:** 3

**Summary:**

This paper introduces SynopticMind, a multimodal large language model (MLLM) designed for weather forecasting report generation. The authors propose a new task, named Weather Forecasting Report (WFR) and construct the first instruction-tuning dataset, WFInstruct, which pairs ERA5-based meteorological heatmaps with expert-written weather reports. Built on Qwen2.5-VL-7B, SynopticMind is trained through a three-stage process combining supervised fine-tuning, rejection sampling, and preference optimization to enhance factual accuracy, lexical diversity, and alignment with human reporting styles. The paper’s main contributions are:

1. Introduction of the WFR task and WFInstruct dataset, enabling multimodal instruction-tuned weather report generation.

2. Development of SynopticMind, the first open-source MLLM specialized for generating human-readable and meteorologically accurate forecasts.

3. Comprehensive evaluation demonstrating that SynopticMind-DPO surpasses GPT-5 and other leading MLLMs in accuracy, detail, and cross-city generalization.

**Strengths:**

The paper is well-written and easy to follow.

The authors propose a novel multimodal task and dataset for weather report generation, extending instruction tuning to a new and practical domain.

The three-stage training pipeline (SFT→RFT→DPO) is methodologically sound and effectively combines existing alignment techniques in a new application setting.

The work demonstrates clear practical significance by outperforming strong baselines and highlighting a promising direction for domain-specialized MLLMs.

**Weaknesses:**

1. While the paper introduces a new dataset and task, the novelty is primarily in dataset construction and application rather than in model architecture or learning principles; the method relies heavily on established techniques (SFT, RFT, DPO) without clear algorithmic innovation.
2. All evaluation depends on LLM-as-a-Judge scoring, which is known to be prone to bias [1]. Complementary human or meteorological expert assessments would improve credibility.
3. My biggest concern is the evaluation design. The LLM-as-a-judge provides only an overall score for entire reports, which may mix correct and incorrect statements without transparency.  A more rigorous and explainable evaluation would be decomposing reports by temporal segments or forecast points, enabling fine-grained comparison with human reports and clearer correctness signals.
4.  "Direct Preferred Optimization" on line 128 should be "Direct Preference Optimization",


[1] Gu, Jiawei, et al. "A survey on llm-as-a-judge." arXiv preprint arXiv:2411.15594 (2024).

**Questions:**

Please refer to weakness.

---

### Official Review · Reviewer_DYHt · 2025-11-01

**Soundness:** 2
**Presentation:** 2
**Contribution:** 2
**Rating:** 2
**Confidence:** 3

**Summary:**

This paper focuses on the task of weather forecast report generation using multimodal large language models (MLLMs). They construct a instruction-tuning dataset for this task (WFInstruct), and then develop the SynopticMind model, which specializes in weather report forecasts. The authors claim that their model surpasses GPT-5 and other SOTA models in this task.

**Strengths:**

(1) They address an important task with a clear gap in the literature where current models are not strong.

(2) Their three stage training pipeline appears effective, with each stage providing additional gains.

(3) They include extensive ablations, such as examining the effects of different variables (Figure 6), which offer useful insights.

**Weaknesses:**

(1) While asking an LLM to assign a score from 0 to 5 is useful, it is too abstract on its own. A complementary approach would be to extract claims from both the ground truth and the generated reports and compute precision, recall, and F1 over those claims.

(2) In line 234, the threshold is set to 5 for HK and 4 for other cities. The rationale for this difference is unclear and appears arbitrary.

(3) Given the abstract nature of the metric, I am concerned about how reliable the claim is that the model outperforms GPT-5 and similar systems.

(4) Lines 351 to 365 state that training on some cities yields good results on geographically close locations but poor results on distant ones. This warrants deeper analysis, as it may indicate that evaluation on nearby locations is too similar to the training distribution and suggests overfitting rather than true generalization.

(5) The paper’s writing needs significant improvement. Table captions are not self explanatory. For example, Table 1 should state that the column headers are city acronyms, and the word performance should be explicitly defined in Table 2. Many explanations appear too late, and readers must search across sections to understand earlier references. The ordering of information is off, with concepts introduced long before they are clearly explained.

**Questions:**

(1) The paper mentions four distance based strategies to measure diversity, but it is unclear how these strategies are combined or selected in practice.

Please also refer to the "Weaknesses" for additional questions.

---

### Official Review · Reviewer_Emta · 2025-11-01

**Soundness:** 3
**Presentation:** 2
**Contribution:** 2
**Rating:** 4
**Confidence:** 3

**Summary:**

This paper presents SynopticMind, a multimodal instruction-tuned LLM for automatic weather report generation. We introduce the Weather Forecasting Report (WFR) task and release WFInstruct, a city-level paired dataset of weather imagery and human reports. Built on Qwen2.5-VL-7B, SynopticMind uses a three-stage training pipeline to improve report quality and is evaluated against leading closed-source LLMs with extensive ablation studies.

**Strengths:**

1. The paper is clearly motivated by the information overload and subjectivity issues in current weather report workflows
2. The introduction and public release of WFInstruct offer a much-needed benchmark for aligning multimodal models with real weather forecast report writing, addressing a major gap in the field.

**Weaknesses:**

1. Limited Breadth of Data/Domains:dataset and results are limited to ten cities, the results may not fully represent global performance or non-English settings, and generalization to low-data regions is not directly evaluated.
2. Evaluation Standardization and LLM-as-Judge Reliability: Main evaluation relies on LLM-as-judge, There is no human expert evaluation reported, and it remains unclear how robust or calibrated LLM-judgment is to nuanced meteorological reporting, which may undermine the conclusiveness of the main results.
3. Recent vision-language models for meteorological (Specialized model) are missing.

**Questions:**

1. Is there any experiment/analysis regarding the robustness of predictions to rare but high-impact severe events (e.g., typhoons, severe storms), versus routine, fine-grained forecasts?
2. Can the authors clarify to what extent the reported improvements in Table 2 would persist if using human (domain expert) evaluators.
3. How difficult would it be to extend WFInstruct to additional languages or locations? Are there any preliminary results or barriers for, say, European or African cities, or non-English reports?

---

### Note · Authors · 2025-11-18

I have read and agree with the venue's withdrawal policy on behalf of myself and my co-authors.